# Recovery of Pure Lead-Tin Alloy from Recycling Spent Lead-Acid Batteries

**DOI:** 10.3390/ma16175882

**Published:** 2023-08-28

**Authors:** Daniel Malecha, Stanisław Małecki, Piotr Jarosz, Remigiusz Kowalik, Piotr Żabiński

**Affiliations:** 1Baterpol S.A., ul. Obr. Westerplatte 108, 40-335 Katowice, Poland; 2Faculty of Non-Ferrous Metals, AGH University of Krakow, al. Mickiewicza 30, 30-059 Krakow, Poland

**Keywords:** recycling, lead acid battery, recovery method, recovery metals, circular economies, refining process, hazardous wastes, lead, tin, aluminium

## Abstract

Spent lead–acid batteries have become the primary raw material for global lead production. In the current lead refining process, the tin oxidizes to slag, making its recovery problematic and expensive. This paper aims to present an innovative method for the fire refining of lead, which enables the retention of tin contained in lead from recycled lead–acid batteries. The proposed method uses aluminium scrap to remove impurities from the lead, virtually leaving all of the tin in it. The results of the conducted experiments indicate the high efficiency of the proposed method, which obtained a pure Pb-Sn alloy. This alloy is an ideal base material for the production of battery grids. This research was carried out on an industrial scale, which confirms the possibility of facile implementation of the method in almost every lead–acid battery recycling plant in the world.

## 1. Introduction

Natural resources of non-ferrous metals are limited, and in many cases, their extraction from deposits is very expensive. Manufactured products containing non-ferrous metals can be hazardous to the environment after the end of their life cycle. A perfect example is lead–acid batteries (LABs), which are used primarily in automotive applications in classic combustion cars and hybrids. They are also used as auxiliary batteries in electric cars to power devices that use 12 V power, e.g., the on-board computer that controls the high-voltage battery, power steering systems, interior lighting, and more [1]. In addition to automotive applications, LABs are used in stationary energy storage, traction batteries, scooters, and electric bicycles. Thanks to their low-cost raw materials, high operational safety and reliability even under extreme temperature conditions, and the maturity of manufacturing and recycling technologies, LAB production has steadily increased in recent decades [2,3]. The primary raw material in LABs is lead, accounting for about 60% of the total battery weight [4].

One of the main technical problems associated with LABs, leading to premature battery failure, is the rupture of their positive current collector. The causes of this phenomenon are, on the one hand, corrosion caused by the electrolyte and, on the other hand, oscillatory mechanical loads resulting from charge–discharge cycles accompanied by changes in the specific volume of the active material in the grid cells [5,6]. As a result, increasingly stringent requirements are being imposed on the anti-corrosion and strength properties of alloys for the production of positive current collectors. The most common materials used for this purpose today are alloys of the ternary system Pb-Ca-Sn, where the tin content ranges from 0.1% to almost 1.5% and the calcium content is around 0.08% [7,8]. This makes tin the second main alloying additive after antimony. Increasing amounts of tin in manufacturing battery grids and components also increases the content of this element in used LABs, providing enormous potential for the recovery of this valuable metal during the recycling process.

There are two distinctly different approaches to processing LABs in the recycling industry. The first involves melting down the charge consisting of whole batteries after removing the electrolyte. This process is conducted in shaft furnaces. From an environmental point of view, this is far less favourable than the second process, which involves breaking up, crushing, and physical separation into individual components [9,10]. This process yields battery paste (30–50%), a metallic fraction from grids and components (20–30%), spent electrolyte (10–22%), polypropylene from casings (5–8%), spacers (4–7%), and scrap iron (about 2% of the battery weight) [11,12]. Lead from the components and battery paste is fed through traditional pyrometallurgical processes. Re-granulate is made from polypropylene, while the electrolytes can be purified or neutralised [13,14]. Battery paste is a mixture of 50–60% PbSO_4_, 15–35% PbO_2_, 5–10% PbO/Pb(OH)_2_, 2–5% metallic Pb, and a small amount of impurities (iron, antimony, tin, barium, and others) [15,16]. Battery paste can not only be processed by pyrometallurgical methods in a rotary furnace after desulphurisation or in a QSL kiln but also by hydrometallurgical methods and combinations thereof. Although these methods are little used in industry, they have considerable potential in terms of energy efficiency and lower emissions of heavy metals into the air [17,18].

Pyrometallurgical processing of the metallic fraction alone from grids and components generates a slag waste containing 3–5 wt% tin [19]. The presence of substantial amounts of sulphides makes the slag produced after LAB recycling processes not the best material for pyrometallurgical processing. A better method of recovering the lead contained in the slag is the hydrometallurgical process. Research is also being conducted on the possibilities of recovering other metals, e.g., iron and tin, from slags using hydrometallurgical processes as well [20,21,22]. As mentioned, the tin comes from battery grids and components. When conducting the classic lead refining method, the tin passes first into the dross and then into the slag. In the lead refining process, such as Harris refining, which consists mainly of oxidation, elements such as tin, antimony, and arsenic are removed from the lead [23]. Transferring the tin to the oxidised dross significantly hinders its potential recovery and increases costs [24]. On the other hand, there is a need to remove impurities such as antimony and arsenic from lead. An alternative way of refining lead is, therefore, proposed, taking into account the removal of harmful impurities without reducing the tin content. This will allow for the optimal use of the tin contained in secondary lead for the production of lead alloys with tin and other additives. A method ‘borrowed’ from the classical processes for refining tin from antimony and arsenic, i.e., using metallic aluminium, should work well for this purpose [23,25].

The addition of aluminium can be used to remove antimony, copper, and arsenic from lead. This process has been well known since at least the previous century [26]. The double system of aluminium and lead does not show the formation of inter-metallic compounds that allow for permanent connections (Figure 1) [27,28]. The same applies to the Al-Sn double system (Figure 2) [27,29]. Thus, as shown in the Al-Pb (Figure 1) and Al-Sn (Figure 2) diagrams, introducing aluminium into a metal bath with a lead–tin alloy at a temperature above 660 °C will cause its complete dissolution. Aluminium introduced into a metal bath will react with antimony. According to the Al-Sb diagram (Figure 3), an intermetallic AlSb compound with a melting point of about 1060 °C will then be formed [27]. A similar situation will take place for arsenic, where, according to the Al-As diagram (Figure 4), an intermetallic AlAs compound with a melting point of about 1750 °C will be formed [27]. Reported melting points of chemical compounds may vary depending on the source. The resulting Al-Sb and Al-As intermetallic compounds are insoluble in lead at the temperature at which the refining process is carried out (maximum of 700 °C). Due to its lower density in relation to lead, the resulting intermetallic compounds will flow out to the surface of the metal.

Research has already been conducted on the use of aluminium to remove antimony and copper from lead from scrap lead–acid batteries in terms of lead pre-refining. To remove antimony, arsenic, and tin more thoroughly, an additional refining process using the anodic polarisation of lead in molten sodium hydroxide has been proposed [30,31,32]. A method of removing nickel from lead using an aluminium additive to form an intermetallic compound is also known [33]. The Al-Ni phase diagram (Figure 5) shows that aluminium with nickel can form various intermetallic compounds, and some of them, e.g., AlNi and AlNi_3_, have a melting point above 1000 °C [27]. The use of an aluminium or aluminium–zinc alloy additive to lead to thoroughly remove copper has also been patented [34]. The mechanism is similar to that described earlier. Aluminium combines with copper to form intermetallic compounds with relatively high melting points, as shown in the Al-Cu phase diagram (Figure 6) [27,35]. The dross formed in this process was tested, and the presence of intermetallic compounds such as Al_9_Cu_11_, Al_2_Cu_3_, and Al_2_Cu was found [36].

An interesting approach to the subject of lead refining is presented by new patents, which propose the use of a mixture consisting of 8–30 wt% aluminium, 40–60 wt% calcium, 2–6 wt% coke, and lead powder (proportions may vary slightly from patent to patent). The result of conducting such a process is a lead–tin alloy containing substantial amounts of calcium. Such a material can be a good base for the production of Pb-Ca-Sn battery alloys [37,38,39,40]. However, the research presented and the patented methods for using aluminium in lead refining have not been widely applied in the industry.

Global annual production of refined lead in 2020 was approximately 11.7 million Mg, of which more than 85% was for the production of lead oxide and lead alloys used in LAB production [12]. For refined tin, its global consumption in 2017 reached 381,000 Mg. This is expected to increase further in the coming years [41]. World tin reserves in 1999 were 9.6 million Mg and had declined to 4.7 million Mg by 2016 [19]. Data from the last decade shows that the annual amount of LAB used and discarded was more than 2.6 million Mg in China, about 1.8 million Mg in the Americas and about 1.5 million Mg in Europe, bringing the total for these regions to 5.9 million Mg [20,42]. Almost 95% of the recycling plants for used LABs relied on pyrometallurgical methods [20], in which, during the oxidation-based refining process, the entire tin content was transferred to the dross.

This paper aims to present a novelty complex method for the pyrorefining of lead, which enables the retention of the tin contained in lead from recycled LABs. The pure lead–tin alloy produced will be an ideal base for the production of Pb-Sn-Ca, Pb-Sn, and Pb-Sn-Se alloys, currently used in the battery industry. According to the authors’ best knowledge, despite the creation of patents for the use of aluminium in lead refining processes, there are no scientific literature data on the comprehensive pyrorefining of lead without the loss of the tin contained within it.

## 2. Materials and Methods

### 2.1. Refining of Secondary Lead Containing Tin—Description of the Process

The method of refining the lead–tin alloy proposed by the authors classically begins with the process of drossing the lead. This process is based on the low solubility of copper, nickel, and cobalt in lead at low temperatures. It is conducted in refining kettles, where the lead is stirred and cooled to a temperature of about 350 °C. The resulting melting loss (known as dross) is then collected from the surface. In addition to copper, nickel, and cobalt, a considerable proportion of arsenic and a smaller proportion of antimony also pass into the dross, as these metals form the difficult-to-melt compounds Cu_5_As_2_ and Cu_3_Sb with copper. If, during the drossing process, a sufficiently low content of copper in lead was not obtained, the next stage of refining is decoppering with sulphur. This process is conducted at a temperature of about 330 °C by adding granular elemental sulphur or galena [33,43].

An essential element aimed at saving added aluminium is the removal of sulphur to less than 0.0001%. Aluminium combines very intensively with sulphur, and therefore, the content of this element in lead may hinder the transition of aluminium to lead and cause substantial amounts of dross [27,44]. Sulphur is removed using sodium hydroxide added in 2 to 3 times the weight of the sulphur still in the bath. Sodium hydroxide is added at 390 °C to 410 °C. The amount of NaOH (in powder form) added must be chosen precisely so as not to cause oxidation of the tin contained in the lead.

According to the classical refining scheme, the next step would be oxidative refining, i.e., the purification of lead from zinc, tin, arsenic, and antimony by blowing oxygen and/or air with a lance at metal temperatures above 600 °C. Another way of removing tin, arsenic and antimony by oxidation is refining based on the so-called Harris method. This involves heating the lead to about 430 °C, and then adding sodium hydroxide NaOH and sodium nitrate NaNO_3_ in powder form. The primary pollutant oxidizing agent is sodium nitrate, which decomposes into Na_2_O, N_2_, and O_2_ already at 300 °C. The oxygen produced from the decomposition of saltpetre is a highly active oxidant, which, according to the law of mass action, oxidises primarily lead, and the only resulting compound Na_2_PbO_2_ combines with tin, arsenic, and antimony as well as zinc, sulphur, and tellurium [45]. In our case, however, to retain all the tin in the lead, the metal is heated to between 640 °C and 700 °C and metallic aluminium is added in an amount of 0.2 to 0.5 times the sum of the amounts of antimony and arsenic impurities by weight. Lead with aluminium is mixed until the aluminium is completely dissolved. The application of aluminium to lead can be conducted by various methods, e.g., by a feeder in which the aluminium is poured directly into a funnel created by stirring the lead (Figure 7a), or by putting the aluminium into a steel basket and dipping it into the lead (Figure 7b). Aluminium in lead effectively combines with elements such as antimony, arsenic, nickel, tellurium, copper, and sulphur to form stable chemical compounds at these temperatures.

The lead is then cooled to between 370 °C and 500 °C, and coke is added in an amount of 0.001 to 0.003 times the weight of the total refined lead–tin alloy to dry the dross formed on the lead surface. This process must be conducted because, otherwise, it would be difficult to separate the resulting aluminium-rich foam from the lead. Once the foam dries, it turns into loose dross, which must be collected mechanically. The last stage is the removal of aluminium residues with the use of caustic soda NaOH (in powder form) according to the amount of aluminium weight in the metal bath times 2, at a temperature of 400 °C to 480 °C. Adding more soda will cause losses in tin, and a smaller amount will not remove all of the aluminium.

The result of the process conducted is a lead–tin alloy with a content of total impurities, such as antimony, arsenic, sulphur, nickel, copper, tellurium, and aluminium, of less than 0.001% by weight. If the lead comes entirely from the LABs or the dross after refining the lead from the LABs, refining operations, such as the removal of silver, bismuth, and thallium, are practically not performed [46,47].

### 2.2. Origin and Characteristics of the Material

Changes in the LAB property requirements over the last few decades have resulted in changes in the composition of lead alloys used for manufacturing battery grids and components. Tin significantly improves the properties of lead–calcium alloys, increasing their tensile strength, creep resistance, and passivation resistance. The corrosion rate of the grid also decreases markedly as the tin content increases to 1.45%. Pb-Ca-Sn alloys are mainly used as positive grids in VRLA-type LABs, while the tin content of these alloys is in the range of 0.1–1.5%. In addition to the positive grid in the battery, tin is also used in lead alloys for top bars (straps) and terminals. The alloys used for these contain tin at 0.8–2.7% levels and may also contain selenium as a nucleating agent to promote uniform grain structure and reduce strap corrosion [48,49,50].

A company located in the European Union dealing with recycling has made available its chemical analyses of lead after the melting of LAB grids and components for the last 13 years. A total of 18,114 chemical analysis results were provided for statistical analysis. Each analysis determined the chemical composition of a cast block of raw lead weighing between 1 and 9 Mg. These blocks were cast from a total of over 0.5 million Mg of used LABs processed in the plant during this period. The recycled LABs were of diverse types (no pre-selection was conducted) and came entirely from within the European Union. Chemical analyses were conducted on an optical emission spectrometer (in 2010 model ARL 3460, then from 2019 ARL 4460, and in 2020, model ARL 8860 from Thermo Fisher Scientific Waltham, MA, USA.). The graph in Figure 8 shows the change in the content of the main alloying additives, i.e., antimony, tin, and arsenic, from year to year. The points on the graph correspond to the average value of a given element in a particular year.

From the correlations shown (Figure 8), an increase in tin content from an average of 0.12% in 2010 to an average of 0.45% in 2022 is clearly noticeable. The trend line shown suggests a further increase in the tin content of the spent LABs. The antimony content has oscillated around a value of 1.2%, while the arsenic value increased until 2015, when it reached an average value of 0.22%, and has been steadily decreasing since then.

The charge for the tests was the raw lead from the recycling of lead–acid batteries and, more specifically, from the melting down of battery grids and components. The use of shorter and longer reduction times (up to 6 h) in the tilt rotary furnace during remelting of the metallic fraction from the LABs resulted in crude lead blocks richer or poorer in tin. This made it possible to select blocks with more than 1% tin as a charge for the refining kettle. In addition to lead derived directly from the recycling of LABs, lead derived indirectly from the processing of LABs, i.e., from the remelting of tin-rich refining dross, was also used. Such dross is formed after classically conducted refining processes, e.g., in air or oxygen oxidation, when mainly tin is removed from lead [24]. This type of material smelting in the tilt rotary furnace yielded crude lead with a tin content of up to more than 12%.

In the process, aluminium scrap was used instead of pure aluminium for economic and environmental reasons. Care was taken to select the chemical composition of the aluminium scrap used accordingly so that the Al content was above 98%. The main contaminant in the aluminium scrap used was iron, and its value varied from 0.1 to 0.8% depending on the batch. The second main contaminant was zinc, which ranged from 0.01 to 0.7%. The elements that also occurred above the decimal part of a percent were silicon, copper, and manganese. Other elements found in the aluminium scrap used were at or below hundredths of a percent. The average chemical composition of the aluminium used in the tests is shown in Table 1.

Based on the preliminary tests conducted, it was determined that a magnesium content of more than 0.1% in the aluminium scrap had a negative effect on the process and, in addition to this, the magnesium present in the aluminium transferred to lead. Elements such as iron, silicon, copper, and manganese were considered to be harmless impurities in aluminium scrap for the process. The transition of these elements from aluminium to lead was also not observed. The form of aluminium also played a significant role. Too fine a fraction may not have passed into the lead due to oxidation, while using a large chunk aluminium fraction would have prolonged the entire process. Therefore, a cube with sides of 5–3 cm, commonly available on the aluminium scrap market, was considered the optimal shape.

### 2.3. Place and Method of Testing

Tests of the proposed alternative method of refining the lead–tin alloy were conducted in industrial conditions at the lead refining department of a lead–acid battery recycling company. The tests were conducted to determine the basic parameters of the process, such as the temperature, refining time, introduced additives, and methods of their application in terms of the efficiency of refining the lead–tin alloy and the degree of tin retention in the lead. The refining division where the tests were conducted has a number of lead refining kettles with a maximum capacity of 110 Mg of lead. Each of the kettles was equipped with a set of gas burners with a total output of 540 kW. The lead temperature in the kettles was measured using two independent systems. Each system was fitted with a type K (NiCr-NiAl) CTK-2000 temperature sensor, accuracy class: 1 (according to IEC 60584-1 [51], IEC 60584-2 [52]). The furnaces had covers with a mixer equipped with a 45 kW motor with infinitely variable control.

### 2.4. Chemical Analysis

The change in the chemical composition of the analysed samples was crucial in the research presented. Each of the lead samples collected was analysed on an ARL™ iSpark 8860 spark excitation optical emission spectrometer and, to confirm the results, also on an older model ARL™ 4460, both spectrometers are from Thermo Fisher Scientific Waltham, MA, USA. Both spectrometers had analytical curves calibrated using certified reference materials in the range tested for each of the elements presented in the results. The differences among the results on the two spectrometers were within the range of measurement error, so only analyses from the newer ARL iSpark 8860 model are presented in the results.

The fed aluminium and the dross collected in the process were analysed for chemical composition using an ICP-OES-iCAP™ 7400 Thermo Fisher Scientific Waltham, MA, USA inductively coupled plasma optical emission spectrometer and an Ultima Expert from Horiba Scientific for ICP-OES verification. Elements such as Pb, Fe, and Zn were determined on an iCE™ 3300 AAS Atomic Absorption Spectrometer from Thermo Fisher Scientific Waltham, MA, USA. As a control, a sample of the dross was subjected to additional analysis by classical methods, through titration, to verify the lead, antimony, arsenic, and sulphur content.

Expanded uncertainty was calculated for the obtained measurements with a coverage factor of k = 2, corresponding to a confidence level of about 95%.

On a JCM-6000PLUS scanning electron microscope from JEOL Tokyo, Japan, microscopic observations of the dross were made along with mapping of the surface distribution of elements and quantitative analyses from the points marked on the image. The tests were conducted using an accelerating voltage of 10 kV.

A sample of the dross was also subjected to X-ray phase analysis to determine the compounds formed. The analysis was performed using a Rigaku Tokyo, Japan, MiniFlex II X-ray diffractometer.

## 3. Results and Discussion

### 3.1. Preliminary Information

The discussion of the results of the tests is focused on the analyses of the changes in the concentrations of antimony, arsenic, and tin. The other components were less important due to their low contents. It should be mentioned that insignificant amounts of arsenic were removed before the antimony, so the arsenic content dropped very quickly to values of a few ppm. A critical issue when refining a Pb-Sn alloy from antimony with aluminium is the sulphur content of the starting alloy. It should be reduced to a content of several ppm before aluminium is introduced. Any amount of sulphur consumes a certain amount of aluminium, forming AlS and Al_2_S_3_ compounds with high melting points [27,44], affecting the wear and tear of the aluminium and significantly increasing the processing time. The phase diagram of the Al-S system is shown in Figure 9.

The tests were conducted during the normal operation of the lead refinery, which made it impossible to conduct the tests under near-lab conditions, mainly due to the logistics of the processes at the refinery. This resulted in an uneven feed of aluminium, which sometimes artificially increased the process times. However, it provided an opportunity to analyse the process under different conditions. A total of eight tests were conducted, and in each of the tests, the mass of lead in the kettle was about 100 Mg. The process of refining the lead–tin alloy with aluminium was conducted in each case according to the scheme shown in Figure 10.

### 3.2. Refining of Alloys with Low Tin Content

The first series of tests was conducted for lead containing tin in amounts similar to the alloys used in battery manufacture. The tin content varied from 1.26% to 1.53%. However, the antimony and arsenic contents differed significantly, varying from 1.02–5.83% and 0.0004–0.188%, respectively. Chemical analyses of the lead used in the research before and after the refining process are presented in Table 2. Figure 11a–d shows the changes in the antimony and tin concentrations during the time the tests were conducted. An additional summary of the changes in the Sb concentrations is shown in Figure 11e. The total contents of impurities from other elements (such as copper, nickel, and tellurium) in Tests 1, 2, and 4 were within the limits allowed for lead–tin alloys and calcium and were very quickly reduced to even lower levels together with antimony and arsenic. The highest total contents of these three contaminants was in the lead in Test 3, where copper was at 725 ppm, nickel was at 76 ppm, and tellurium was at 13.9 ppm. Figure 11f shows the changes in the concentrations of these elements together with arsenic, which was 158 ppm. The rate of removal of these elements with aluminium was high, and to accurately illustrate the changes in the concentrations over time, a logarithmic scale was used in the graph. Even the first addition of aluminium caused a dramatic decrease in these elements. Ultimately, the arsenic, nickel and tellurium contents were reduced to below 1 ppm, while the copper was reduced to 28 ppm.

The analysis of the presented relationships shows that regardless of the initial content of antimony and arsenic, the time of the refining process was similar and about 7–8 h. This leads to the conclusion that the removal rate of antimony and arsenic increases with an increase in their contents in lead. This is confirmed by the relationships summarised in Figure 11e. Using the data from these relationships, Figure 12 shows the initial rate of antimony removal after a time of 1 h. With some approximation, it can be said that the rate of this process increased linearly as a function of the antimony content of the starting lead alloy. This was particularly evident in Test 1, where all the aluminium needed for refining was added at the start. This type of relationship is certainly a simplification in this case, as the entire amount of aluminium was not added at the beginning of each test. This inaccuracy was partly eliminated by analysing only the initial short refining period (up to 1 h). The refining tests carried out also prove that aluminium can be added in portions. When dosed at specific times, this does not extend the refining time. Figure 11a–d shows that different Al/(Sb+As) ratios were used, which did not affect the final refining result. The different amounts of aluminium used may result from the way it is fed to the molten metal, which affects its oxidation. With the dosing method mastered, it can be considered that the Al/(Sb+As) ratio should be below 0.3. The relationships shown in Figure 11e,f indicate that the arsenic was removed in the first period of the process, and only later was antimony removed. The chemical analyses of the lead taken during the refining process confirm that, in addition to the arsenic, the copper, nickel, and tellurium were removed during the initial period. Most importantly, however, is the fact that tin remained in lead, which is the basic premise of refining carried out in this way.

It should be noted that the refining tests implemented were conducted until the antimony was removed to a level of approximately 0.05%. During this time, the concentrations of arsenic, copper, nickel, and tellurium in the lead dropped to less than 2 ppm. Once the temperature was lowered, the dross was collected, and the residual aluminium was removed with NaOH in powder form. The final process step was modifying the Pb-Ca-Sn alloy by calcium addition and tin correction. The addition of calcium resulted in a further reduction of the antimony content to a few ppm.

### 3.3. Refining of Alloys with High Tin Content

A second series of refining tests was carried out for lead alloys containing 6.7–8.0% tin and varying amounts of antimony and arsenic, 1.28–7.63% and 0.0006–0.407%, respectively. Chemical analyses of the lead used in the “pre-alloy” tests, before and after the refining process are presented in Table 3. Figure 13a–d shows the changes in the antimony and tin concentrations during the time the tests were conducted. An additional summary of the changes in the Sb concentrations is shown in Figure 13e. In the first three tests, aluminium was added in small portions and at different time intervals, which significantly increased the refining time. It can be seen from the relationships shown that the rate of change in the antimony concentration depended on the amount of aluminium added. Test 4 showed that by adding aluminium in larger portions and more frequently, the refining process could be finished in 6 h. The other process parameters were those obtained when refining alloys with a lower tin content.

As with the low-tin-content tests, in the case of pre-alloys, the one with the highest total amount of impurities (such as arsenic, copper, nickel, and tellurium) was selected, and the changes in their contents during the process is shown in the graph (Figure 13f). In the example shown for ‘Pre-Alloy Test 2’, the baseline content of arsenic was 4071 ppm, copper was 1143 ppm, nickel was 486 ppm and tellurium was 88 ppm. Also in this case, the removal rate of these elements using aluminium was high, and a logarithmic scale is used to illustrate the process better. Ultimately, the arsenic, nickel and tellurium contents were reduced to below 1 ppm, while the copper was reduced to 39 ppm.

It is important to emphasise the fact that, irrespective of the level of lead impurities, it is possible to carry out refining with aluminium in a way that guarantees the removal of Sb, As, Ni, Cu, and Te to very low levels while maintaining virtually constant levels of tin in the lead. This has a significant impact on the economic effect of the refining and alloying processes, as there is no need to add pure tin, and only the concentration of tin is adjusted.

### 3.4. Dross Resulting from the Refining Process with Aluminium

Once the refining process is complete, the dross is collected. Its amount depends on the amount of impurities removed and the amount of lead added. In the presented tests, it ranged between 8 and 25% of the weight of the alloy to be refined. It contained, on average, 47.9% Pb, 34.0% Sb, 9.4% Al, 2.9% Sn, and insignificant amounts of other impurities, as shown in Table 4. It should be mentioned that, in the initial period, the dross contained higher amounts of lead, copper, nickel, and arsenic. As the refining progressed, the antimony and aluminium contents increased, and the Pb, Cu, Ni, and As contents decreased. After refining, all dross was sent for processing to recover the contained metals (Pb, Sb, Sn).

XRD phase analysis was performed and is shown in Figure 14. In the presented phase diagram, it was possible to identify almost all the main peaks. This confirmed the presence of the expected phases. The peaks for lead, intermetallic compound AlSb, lead oxide, and antimony were characterized by the highest intensity. The lowest recorded intensities were for the PbSb compound. The phase analysis performed coincides with the chemical analysis of dross presented in Table 4.

Confirmation of the phases present was also obtained by performing microscopic observations in combination with spot analysis (Figure 15a) and elemental mapping (Figure 15b). The determination of the formation of the AlSb phase is critical. The microscopic image shows grains of exemplary composition: 53.30% Sb, 17.71% Pb, 14.95% Al. The tin present in the dross was a result of its solubility in lead, which passed into the dross mechanically. Grains of oxidised aluminium were also found (Figure 15a). One of the lumps visible in Figure 15a (around point “2”) was enlarged, and the surface was mapped to determine the occurrence of elements. The obtained results clearly indicate the presence of aluminium and elements such as Sb, As, Ni, and Cu (Figure 15b).

## 4. Conclusions

The work reported in the present paper has clearly demonstrated that refining lead from antimony, arsenic, copper, nickel, and tellurium with aluminium produces an outstanding result, and most importantly, virtually all the tin remains in the lead.

Impurities can be removed to levels of a dozen or even less than 1 ppm. Copper is the most challenging and slowest element to remove, so it is best removed by standard methods, i.e., by granulated sulphur, which is also a more economical approach.

The removed impurities form intermetallic compounds with aluminium.

The amount of aluminium added was 120–125% of the stoichiometric requirement. Aluminium should be introduced at a temperature of about 650 °C, and the dross collected at a temperature of about 350–370 °C.

For refining with aluminium, equipment commonly available in lead refining departments, i.e., steel baskets for dissolving Ca-Al or Se mortars, can be used.

The proposed method of refining secondary lead offers the possibility of using the tin already contained in the lead alloy, significantly reducing its consumption for the production of lead alloys for the battery industry.

Instead of pure aluminium, aluminium scrap can be used successfully, provided its composition is controlled. Obtaining aluminium scrap is also much simpler than preparing mixtures with aluminium, lime, coke, and lead in appropriate proportions, which is proposed by patents [37,38,39,40].

During the tests carried out, there was no negative effect of aluminium dissolved in lead on the increase in the wear rate of the steel components of the equipment used, e.g., refining kettles, mixers, etc. On the other hand, the content of aluminium in the dross formed in the process at the level of 7 to 12% forced the appropriate adjustment of their remelting technology.

The costs of lead refining using aluminium scrap are higher compared to other classical lead refining methods [23]. From an economic point of view, this new refining method is profitable to use when there is a significant amount of tin in the lead and relatively low contents of other impurities. These conditions are ideally met by lead from recycled lead–acid battery grids (Figure 8).

Based on the data collected [19,42], it can be estimated that the current volume of recycled batteries within China, the Americas, and Europe reaches more than 5.9 million Mg per year. Assuming a minimum rate of lead recovery from the battery grids and components compacted in them at the level of 20% of the LAB mass and the average content of tin in lead from the remelting of these types of materials at the level of 0.45% (Figure 8), this gives the potential to save over 0.5 million tons of tin per year.

## 5. Patents

Szyndler T., Gnida R., Malecha D., Małecki S., and Jarosz P. Patent Application: A method of recovering lead and tin from recycled batteries or lead dross, application number: PL439991, 2021.

Szyndler T., Gnida R., Malecha D., Małecki S., and Jarosz P. Patent Application: Method of refining a lead–tin alloy, application number: EP23157067.2, 2023.

## Figures and Tables

**Figure 1 materials-16-05882-f001:**
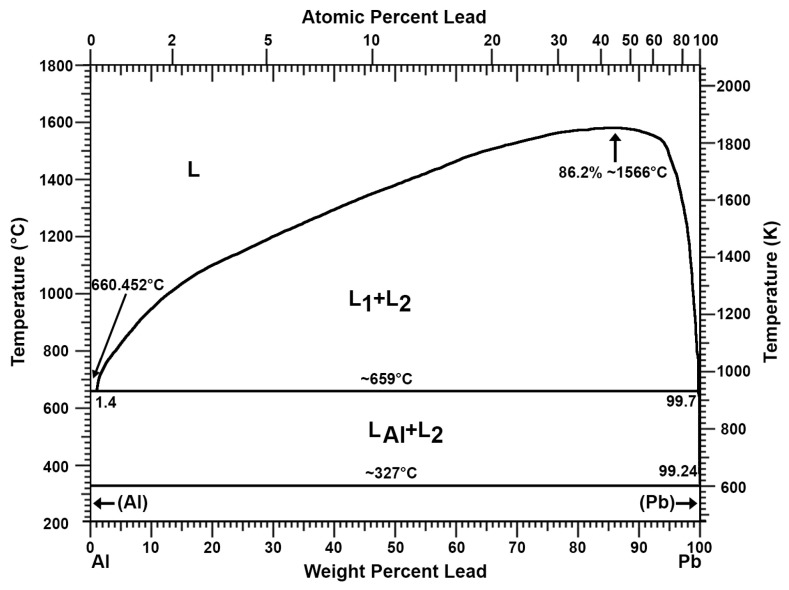
Al-Pb phase diagram [27,28].

**Figure 2 materials-16-05882-f002:**
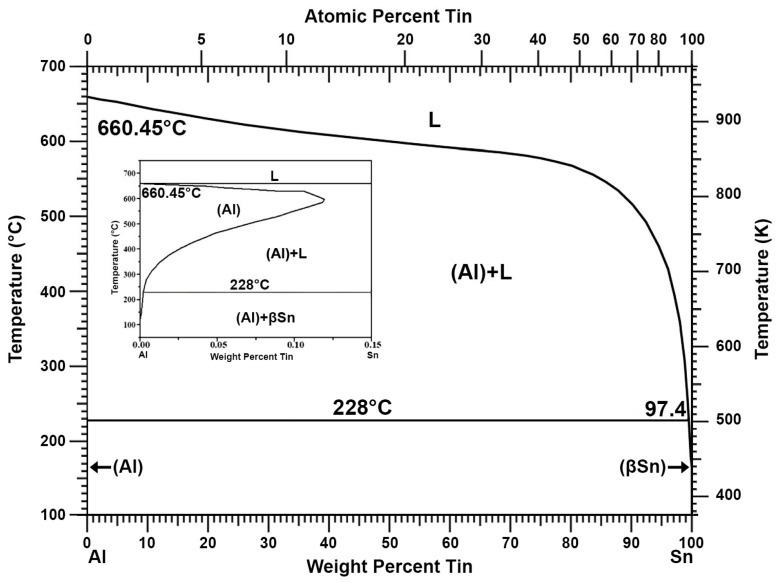
Al-Sn phase diagram [27,29].

**Figure 3 materials-16-05882-f003:**
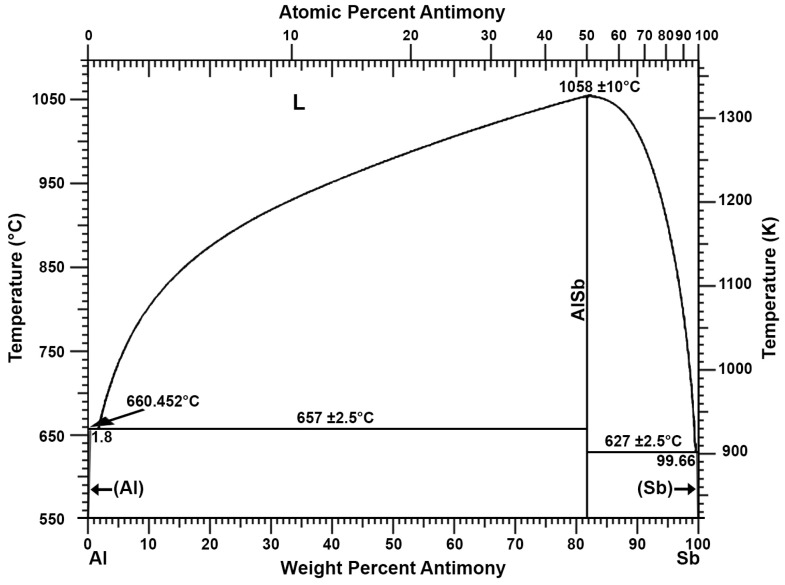
Al-Sb phase diagram [27].

**Figure 4 materials-16-05882-f004:**
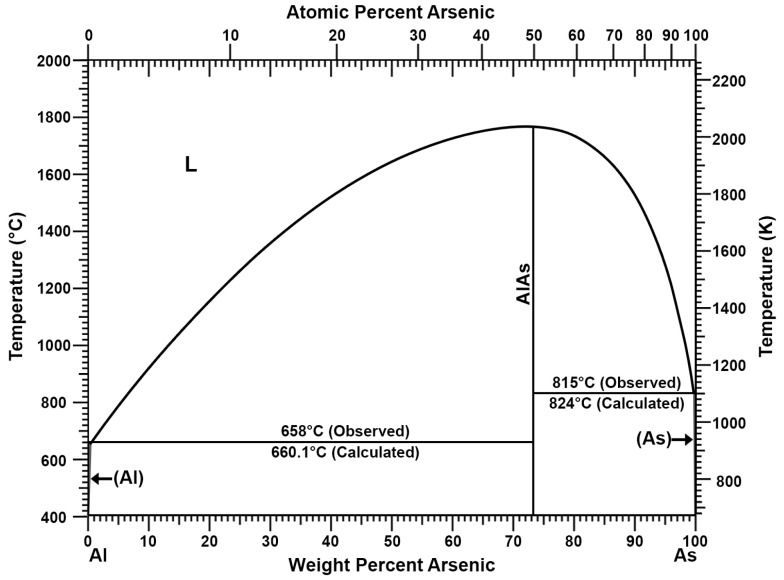
Al-As phase diagram [27].

**Figure 5 materials-16-05882-f005:**
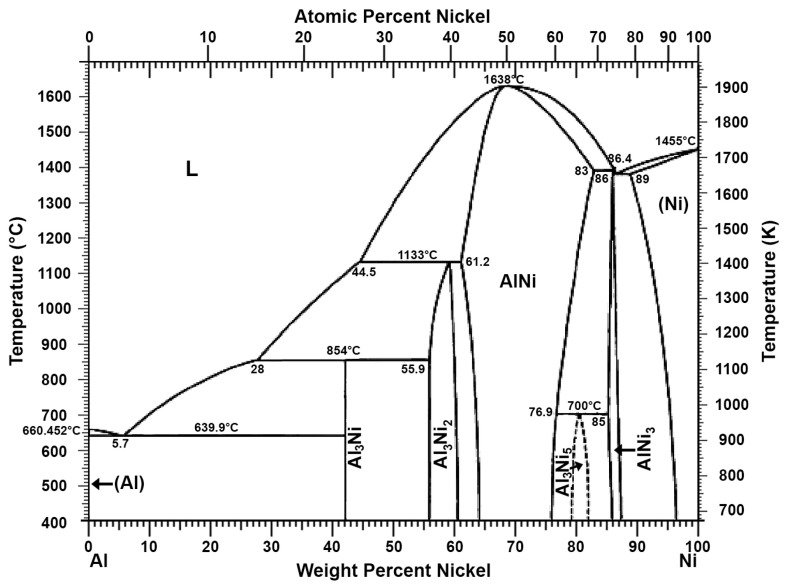
Al-Ni phase diagram [27].

**Figure 6 materials-16-05882-f006:**
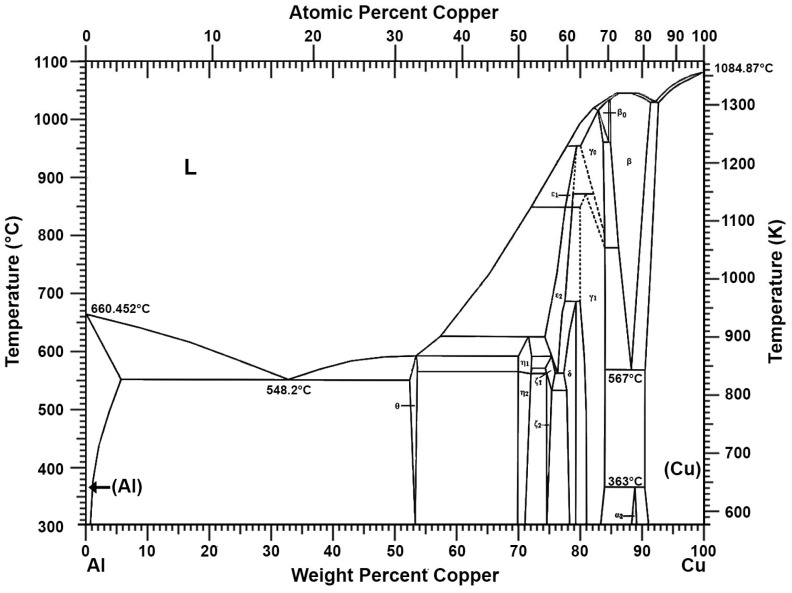
Al-Cu phase diagram [27,35].

**Figure 7 materials-16-05882-f007:**
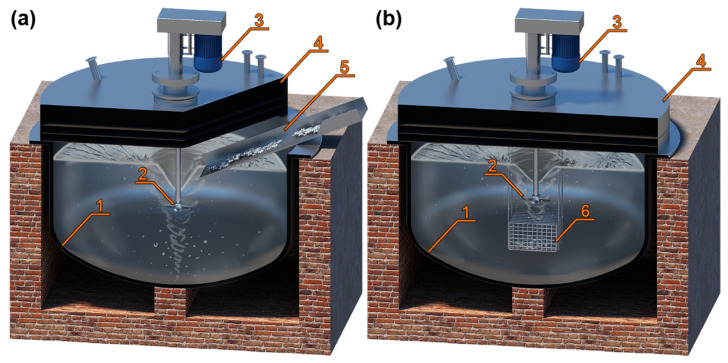
Methods of applying aluminium to lead: (**a**) feeder in which the aluminium is poured directly into a funnel created by stirring the lead; (**b**) putting the aluminium into a steel basket and dipping it into the lead. Description of the components: (1) steel kettle with a capacity of over 100 Mg of lead, (2) agitator propeller, (3) agitator motor, (4) steel cover with technological elements and holes for temperature sensor, (5) feeder in which the aluminium is poured directly into a funnel created by stirring the lead, (6) steel basket with aluminium inside.

**Figure 8 materials-16-05882-f008:**
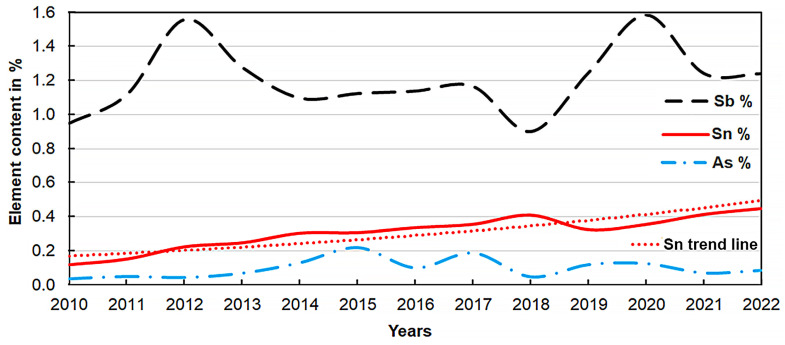
Graph of the change in the content of alloying additives contained in metallic scrap after LAB recycling from 2010 to 2022.

**Figure 9 materials-16-05882-f009:**
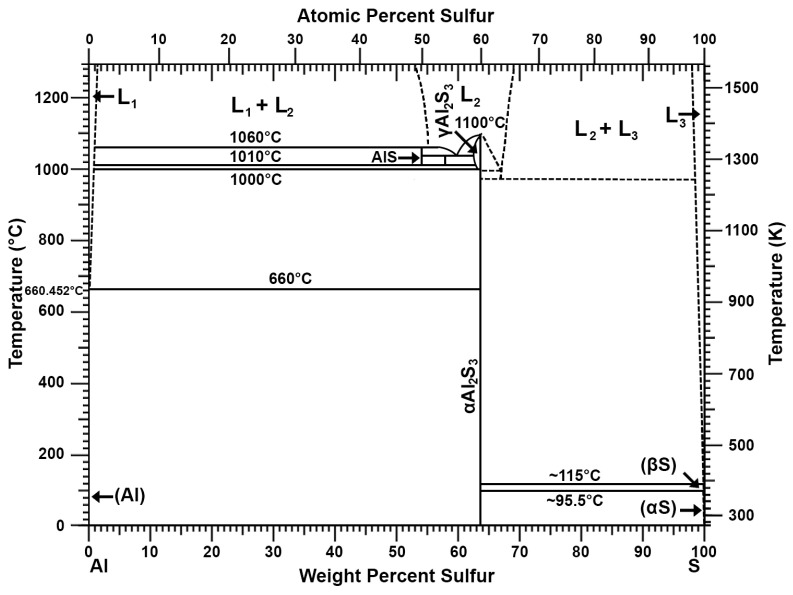
Al-S phase diagram [27,44].

**Figure 10 materials-16-05882-f010:**
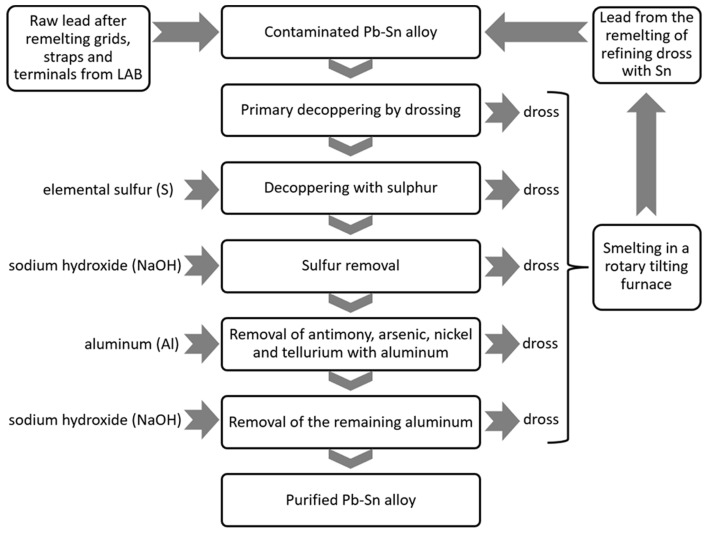
Technological scheme of the Pb-Sn alloy refining process.

**Figure 11 materials-16-05882-f011:**
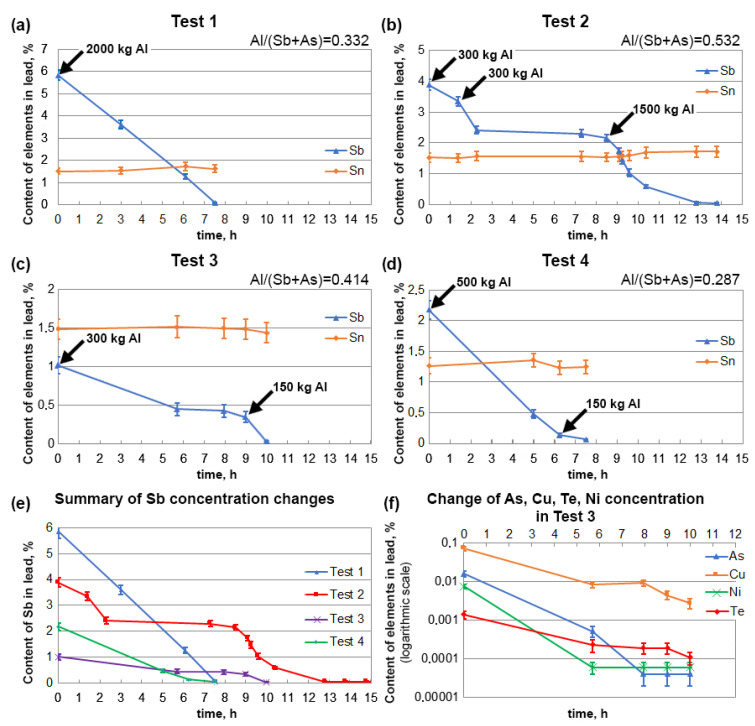
(**a**) Changes in the concentration of Sb and Sn during the test 1. (**b**) Changes in the concentration of Sb and Sn during the test 2. (**c**) Changes in the concentration of Sb and Sn during the test 3. (**d**) Changes in the concentration of Sb and Sn during the test 4. (**e**) Summary of changes in Sb concentrations in tests 1-4. (**f**) Changes in the concentration of As, Cu, Ni and Te during the test 3.

**Figure 12 materials-16-05882-f012:**
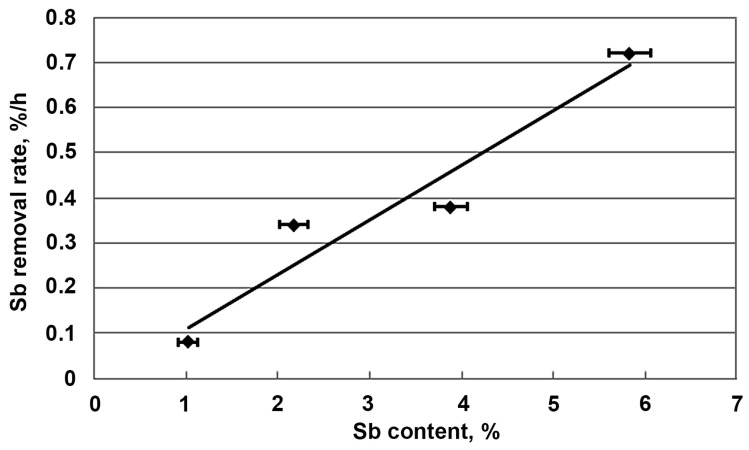
Dependence of antimony removal rate on its initial content in lead during the initial refining period.

**Figure 13 materials-16-05882-f013:**
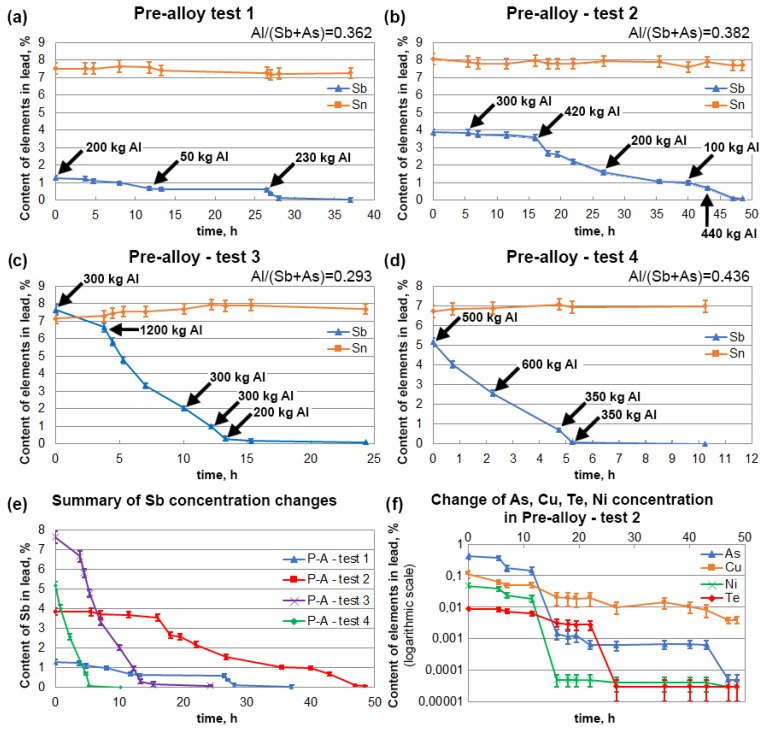
(**a**) Changes in the concentration of Sb and Sn during the pre-alloy test 1. (**b**) Changes in the concentration of Sb and Sn during the pre-alloy test 2. (**c**) Changes in the concentration of Sb and Sn during the pre-alloy test 3. (**d**) Changes in the concentration of Sb and Sn during the pre-alloy test 4. (**e**) Summary of changes in Sb concentrations in pre-alloy tests 1-4. (**f**) Changes in the concentration of As, Cu, Ni and Te during the pre-alloy test 2.

**Figure 14 materials-16-05882-f014:**
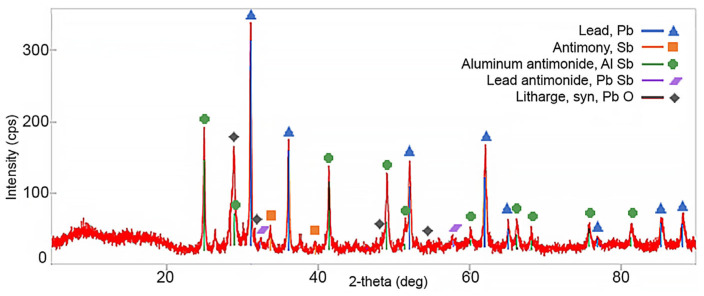
Phase analysis of dross after refining.

**Figure 15 materials-16-05882-f015:**
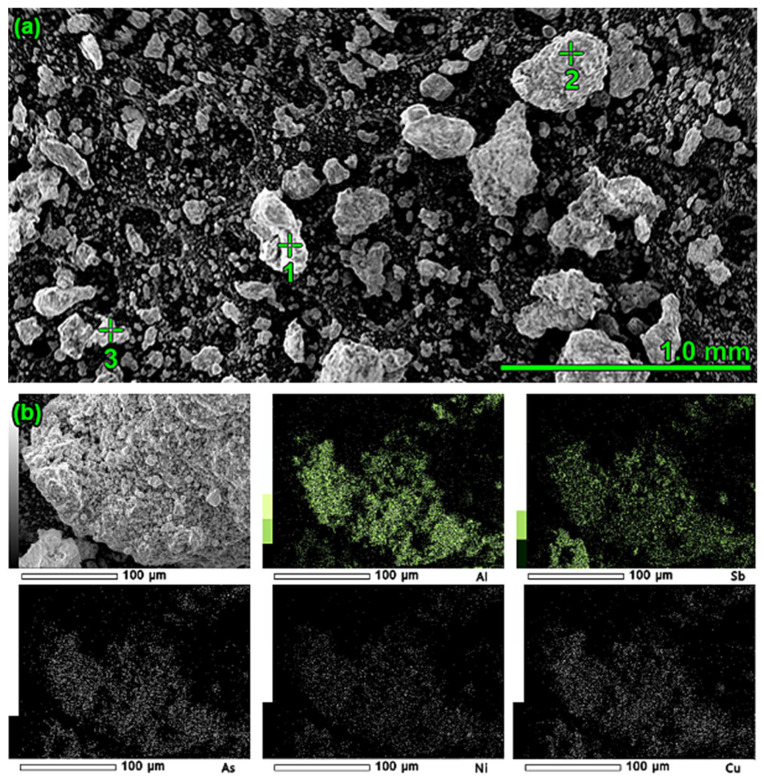
(**a**) Quantitative spot analysis of dross after refining: Point 1: 42.23% Al, 57.77% O; Point 2: 73.45% Pb, 14.19% O, 6.48% Sn, 4.49% Al; Point 3: 53.30% Sb, 17.71% Pb, 14.95% Al, 13.36% O, 0.68% Mg. (**b**) Surface mapping to verify the presence of elements such as Al, Sb, As, Ni, and Cu.

**Table 1 materials-16-05882-t001:** The average value of the chemical composition of aluminium scrap.

Description	Al%	Fe%	Zn%	Si%	Cu%	Cr%	Sn%
Aluminium scrap	98.1 ± 1.8	0.50 ± 0.09	0.49 ± 0.09	0.39 ± 0.06	0.18 ± 0.04	0.028 ± 0.009	0.008 ± 0.003

**Table 2 materials-16-05882-t002:** Chemical analyses of the lead used in the series of tests before and after the refining process.

Number	Desc.	Sb%	As%	Sn%	Cu%	Ni%	Te%
Test 1	Before	5.83 ± 0.23	0.188 ± 0.024	1.51 ± 0.14	0.012 ± 0.003	0.0005 ± 0.0001	0.0007 ± 0.0001
	After	0.096 ± 0.007	0.0003 ± 0.0001	1.61 ± 0.17	0.0021 ± 0.0005	0.00003 ± 0.00002	0.00025 ± 0.00005
Test 2	Before	3.88 ± 0.18	0.062 ± 0.005	1.53 ± 0.14	0.017 ± 0.003	0.0024 ± 0.0003	0.0007 ± 0.0001
	After	0.0097 ± 0.0004	0.0002 ± 0.0001	1.66 ± 0.18	0.0016 ± 0.0004	0.00003 ± 0.00002	0.00007 ± 0. 00002
Test 3	Before	1.02 ± 0.11	0.016 ± 0.003	1.48 ± 0.13	0.073 ± 0.009	0.0076 ± 0.0009	0.0014 ± 0.0003
	After	0.030 ± 0.009	0.00004 ± 0.00002	1.42 ± 0.13	0.0028 ± 0.0008	0.00006 ± 0.00002	0.00011 ± 0.00006
Test 4	Before	2.18 ± 0.15	0.0004 ± 0.0001	1.26 ± 0.11	0.032 ± 0.006	0.00013 ± 0.00006	0.0010 ± 0.0002
	After	0.062 ± 0.005	0.00013 ± 0.00009	1.23 ± 0.11	0.0025 ± 0.0007	0.00005 ± 0.00002	0.0004 ± 0.00007

**Table 3 materials-16-05882-t003:** Chemical analyses of the lead used in the second series of tests before and after the process.

Number	Desc.	Sb%	As%	Sn%	Cu%	Ni%	Te%
P-A Test 1	Before	1.28 ± 0.13	0.038 ± 0.004	7.53 ± 0.37	0.065 ± 0.009	0.0004 ± 0.0001	0.0006 ± 0.0001
	After	0.0082 ± 0.0008	0.00004 ± 0.00002	7.35 ± 0.33	0.0023 ± 0.0005	0.00002 ± 0.00002	0.00009 ± 0.00003
P-A Test 2	Before	3.84 ± 0.18	0.407 ± 0.072	8.04 ± 0.39	0.11 ± 0.03	0.049 ± 0.007	0.0088 ± 0.0012
	After	0.063 ± 0.007	0.00005 ± 0.00002	7.64 ± 0.34	0.0039 ± 0.0009	0.00003 ± 0.00002	0.00003 ± 0.00002
P-A Test 3	Before	7.63 ± 0.29	0.205 ± 0.038	7.15 ± 0.32	0.14 ± 0.04	0.0089 ± 0.0008	0.0018 ± 0.0003
	After	0.091 ± 0.009	0.00008 ± 0.00003	7.69 ± 0.34	0.0021 ± 0.0005	0.00002 ± 0.00002	0.00017 ± 0.00007
P-A Test 4	Before	5.16 ± 0.23	0.0006 ± 0.0001	6.73 ± 0.31	0.025 ± 0.005	0.0005 ± 0.0001	0.0008 ± 0.0002
	After	0.0056 ± 0.0005	0.00003 ± 0.00002	6.98 ± 0.31	0.0015 ± 0.0004	0.00002 ± 0.00002	0.00014 ± 0.00005

**Table 4 materials-16-05882-t004:** Chemical analyses of dross formed in the process (average values from all tests). The row labelled ‘First’ represents the average values of the first collected dross, while the row labelled ‘Second’ presents the average values of the last collected dross.

	Pb%	Sb%	Al%	Sn%	As%	Cu%	Fe%	Ni%	Mg%	Si%
First	58.9 ± 4.3	23.3 ± 2.1	6.83 ± 0.57	3.30 ± 0.38	0.33 ± 0.09	0.22 ± 0.07	0.17 ± 0.04	0.073 ± 0.014	0.04 ± 0.01	0.007 ± 0.001
Second	36.9 ± 3.1	44.6 ± 4.2	12.0 ± 1.1	2.43 ± 0.31	0.02 ± 0.01	0.08 ± 0.02	0.20 ± 0.05	0.006 ± 0.002	0.05 ± 0.01	0.12 ± 0.09
Average	47.9	33.95	9.42	2.87	0.18	0.15	0.19	0.040	0.045	0.064

## Data Availability

The data presented in this study are available upon request from the corresponding author.

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
