# Peer review of "Recovery of Pure Lead-Tin Alloy from Recycling Spent Lead-Acid Batteries"

_materials, 2023, doi:10.3390/ma16175882_

Round 1

Reviewer 1 Report

The manuscript presented by Malecha et al. considers the recovery of le lead-tin alloy from sent LAB. The work is scientifically sound. However, there are some changes that need to be conducted before it can be accepted for publication in Materials J. Here are the details of necessary revision:

1. There are many grammatical mistakes. Please check the manuscript for grammar and English carefully.

2. The aim and novelty of the study is not presented clearly.

3. The abstract needs to be written in a more organized way, indicating the results.

4. Move Fig 1-3 to supplementary file.

5. L 131: space needed 350°C and check for the whole manuscript.

6. What is the molarity of NaOH used?

7. How many times the experiments were performed?

8. Move Fig. 5 to results and discussion section

9. More comments are needed for XRD analysis.

10. Remove numbering from conclusions.

Moderate editing of English language required

Author Response

Thank you very much for taking the time to read our manuscript and provide your comments.

We will address each remark in turn.

  1. The manuscript was corrected by a native speaker.
  2. The description of the aim and novelty of the study has been corrected.
  3. The abstract has been corrected.
  4. We debated for a long time whether to add phase diagrams to the article at all. But we felt the article would be less understandable without them.The truth is that few readers of the article look at the supplement. Adding diagrams to an article also helps to find the article in a search engine. In addition, in the other reviews there was a comment about improving the readability of the phase diagrams (which was done by separating and enlarging them) and in the next review there was a comment about a more accurate scientific explanation of the use of aluminum based on the presented phase diagrams (which was also done). Therefore, please consider whether it is not a good idea to leave the diagram directly in the manuscript.
  5. Spaces have been added.
  6. NaOH was added in powder form. This may not have been explained in the text. So the relevant information has been added.
  7. The industrial-scale experiment was carried out eight times.Clear information on this subject has been added to the text.
  8. Done.
  9. Added more comments for XRD analysis.
  10. Done.

Reviewer 2 Report

Reviewer’s report:

Title: Recovery of pure lead-tin alloy from recycling spent lead-acid battery.

In this work, authors reported the technology to recovery of pure lead-tin alloy from recycling spent lead- acid battery. The manuscript is good in term of topic, presentation, and discussion which may useful for other researchers who work in the field.

1. The labels and specific information on the figures (Figs. 1, 2, and 3) are too small and unreadable.

2. It will be advantageous to include dimensions and other explanations of the construction's component sections at Fig. 4.

Author Response

Thank you very much for taking the time to read our manuscript and provide your comments.

As for the first remark - improved the labels and details in Figures 1, 2, and 3 by separating and enlarging them.

As for the second remark, we introduced the following changes - in Fig. 4 (currently Fig. 7), indicators with numbers have been added, and information explaining what a given element represents has been introduced in the description of the drawing. However, data on the specific dimensions of individual elements was not entered because there was no consent of the management board of the company where the tests were carried out.

Reviewer 3 Report

In this manuscript, the authors proposed a new method to remove impurities from the lead and recovery pure lead-tin alloy from the spent lead-acid battery. In the current lead refining process, the tin oxidizes to dross, making its recovery problematic and expensive. The authors proposed to use aluminum scrap to remove the impurities. The proposed method reduces the need to use tin from primary raw materials. The conducted results also indicate the high efficiency of the proposed method. As a result, a refined Pb-Sn alloy is obtained, which is an ideal base material for producing alloys for the battery industry.

Overall, the manuscript is logically organized and well written. However, there are still several issues that have to be addressed.

In the introduction part, although the authors showed the phase diagrams of several Al-based alloys, yet they didn’t explain the underlying science for using Al to remove the impurities. It is suggested that the authors explicitly explain the reasons for using Al to remove the impurities.

Are there any advantages or disadvantages of this method in recovering lead-tin alloy from the spent LAB batteries compared with other methods?

Author Response

Thank you very much for taking the time to read our manuscript and provide your comments.

As for the first remark -  based on the presented phase diagrams, a few additional sentences have been added explaining in more detail the scientific basis for the use of aluminium to remove impurities such as Sb, As, Ni, and Cu from lead.

As for the second remark, we introduced the following changes - in the first conclusion, the main advantage of the presented method was emphasized, i.e. leaving practically all the tin in the lead. In addition, a conclusion was added stating the disadvantages, which are, among others, the higher cost of refining with the method presented in relation to classical solutions.